# The Role of Nutrition in the Nursing Management of Pressure Ulcers in Adult Community Settings: A Systematic Review Protocol

**DOI:** 10.3390/diseases12100253

**Published:** 2024-10-14

**Authors:** Giovanni Cangelosi, Sara Morales Palomares, Marco Sguanci, Federico Biondini, Francesco Sacchini, Stefano Mancin, Fabio Petrelli

**Affiliations:** 1Unit of Diabetology, Asur Marche—Area Vasta 4 Fermo, 63900 Fermo, Italy; giovanni.cangelosi@virgilio.it; 2Department of Pharmacy, Health and Nutritional Sciences (DFSSN), University of Calabria, 87036 Rende, Italy; sara.morales@unical.it; 3A.O. Polyclinic San Martino Hospital, Largo R. Benzi 10, 16132 Genova, Italy; 4Units of Psychiatry, Ast Fermo, 63900 Fermo, Italy; federico.biondini@sanita.marche.it; 5Polytecnic University of Ancona, 60121 Ancona, Italy; francescosacchini@libero.it; 6IRCCS Humanitas Research Hospital via Manzoni 56, 20089 Rozzano, Italy; stefano.mancin@humanitas.it; 7School of Pharmacy, Polo Medicina Sperimentale e Sanità Pubblica “Stefania Scuri”, 62032 Camerino, Italy; fabio.petrelli@unicam.it

**Keywords:** nutrition, nursing management, pressure ulcers, protocol, frail population, community settings, public health

## Abstract

Background: The aging population drives a growing demand for care, particularly in Europe. It is estimated that approximately 1.5–2 million individuals have a chronic wound. Among these, pressure ulcers (PUs) are one of the most prevalent complications in vulnerable individuals. Malnutrition is a primary risk factor, yet it can be mitigated through proper nutrition and adequate community support. The community nurse plays a crucial role in managing chronic conditions and nutrition through constant and professional monitoring. Aim: This article presents a comprehensive systematic review (SR) protocol to examine the role of community nursing of nutritional intervention of frail population with wound care. Methods: A SR will be conducted according to international standards and reported following the PRISMA Guidelines for SRs. The search will be conducted in PubMed/Medline, Scopus, Embase, and CINAHL, supplemented by grey literature sources. The methodological quality and risk of bias will be assessed using the Critical Appraisal Skills Programme (CASP) framework. The protocol has been registered in the Open Science Framework (OSF). Conclusions: It is anticipated that the findings of this SR will provide new evidence on the relationships between nutritional nursing interventions and wound care management primarily in the community setting.

## 1. Introduction

According to the World Health Organization (WHO), by 2050, 80% of the elderly population will live in low- and middle-income countries. Population aging is accelerating compared to the past, with the number of people over 60 surpassing children under 5 as early as 2020. Between 2015 and 2050, the percentage of people over 60 will double, reaching 22%. The major challenges for the elderly population include hearing problems, chronic diseases, and geriatric syndromes. The WHO promotes healthy aging through policies that combat ageism and by developing supportive communities that ensure integrated and quality care [1,2]. In this context, Italy is one of the countries with the highest aging rates globally, ranking third in Europe and fifth worldwide in terms of average population age at 48.4 years [Central Intelligence Agency (CIA) report 2024] [3]. Additionally, over 14 million individuals in Italy are over 75 years old, with a higher prevalence among females [National Institute of Statistics (ISTAT) report 2024] [4]. This necessarily leads to an increase in care needs due to aging, comorbidities, and bedridden conditions, resulting in higher healthcare costs and challenges in ensuring essential levels of care (LEA) [5]. A possible strategy is the prevention of chronic diseases and complications through specific health education programs led by nurses [6,7]. In this context, aging and chronic diseases increase frailty in the elderly, consequently reducing physical activity, self-efficacy, and overall quality of life (QoL). Physical activity can therefore mitigate these negative effects, which could otherwise impact frail individuals [8]. Physical inactivity and a sedentary lifestyle worsen the functional capacity of the elderly, increasing frailty and healthcare costs. Specific physical activity programs improve functional capacity, cognition, and mood in the elderly, preserving cognitive and social capabilities over time [9,10].

One of the common complications in frail and debilitated individuals is the development of pressure ulcers (PUs), which are very frequent in non-self-sufficient elderly individuals [11,12]. In Europe, it is estimated that around 1.5–2 million people suffer from chronic wounds, accounting for 3% of all healthcare expenditures, primarily due to the time required for care, dressings, and prolonged hospitalization [13]. To address modern challenges in wound management, it is crucial to adopt integrated approaches that combine best clinical practices with cutting-edge technological solutions [14]. The management and prevention of PUs are crucial topics in the care of the elderly and patients with chronic wounds in community settings. Home-based training for caregivers has been shown to significantly improve their and patients’ wound management skills, leading to fewer complications and an overall improvement in the health status of frail patients [15]. PUs remain one of the major negative outcomes associated with nursing care (Nursing Sensitive Outcomes; NSO) [16,17], and there are still few community programs to identify risks early, manage them properly, and, most importantly, prevent them [18,19]. The management and prevention of PUs are crucial in various clinical settings, such as general medicine units or community settings like long-term care. 

Several studies have shown that the ability of nursing staff to use appropriate assessment and prevention tools can better meet patients’ needs both in hospitals and at home [20,21,22]. One of the main risk factors for the development of PUs in the elderly, in addition to cognitive decline and low physical and emotional activity, is malnutrition. Malnutrition in the elderly is often characterized by an inadequate intake of calories, proteins, and micronutrients, which are essential for maintaining skin integrity and overall health. Conditions such as protein-energy malnutrition, vitamin deficiencies (especially vitamins C and E), and dehydration significantly compromise the skin’s ability to repair itself and withstand pressure, making these individuals more susceptible to PUs [23]. Consequently, addressing malnutrition requires a comprehensive approach, including the development of personalized nutritional plans that meet the specific needs of each patient. By proactively addressing malnutrition and the associated risks, healthcare professionals can significantly reduce the incidence of PUs in a fragile population, leading to favorable clinical outcomes and an improved QoL for the elderly in community care settings [24,25].

The importance of this protocol is underscored by the scarcity of studies on nursing management of nutrition to prevent PUs in community settings.

## 2. Materials and Methods

### 2.1. Review Design

To maintain a methodological approach and ensure the significance of the included studies, the systematic review (SR) protocol entails the initial creation of a research protocol. This method is designed to confirm that the selected articles are consistent with the SR’s objectives. The summarization of the data gathered should be clear and thorough, offering a wide-ranging and credible overview for the relevant scientific community. To this end, this SR will adhere to the PRISMA Guidelines [26,27] and the Cochrane Handbook for SRs [28].

### 2.2. Protocol Registration

The protocol for this SR has been recorded in the Open Science Framework (OSF) Register database [29].

### 2.3. Study Objectives and Research Question

The main aim of this study is to create a detailed SR protocol that concentrates on the impact of nutrition in the nursing care of PUs within community environments. Furthermore, the protocol intends to methodically define the approaches for assessing and analyzing the outcomes highlighted in the selected studies. This SR aspires to address the following research questions:


*Primary Research Question:*


What is the effect of various nutritional strategies in the nursing management of adults with PUs in community settings?


*Secondary Research Questions:*


Which nutritional approaches are most effective in improving clinical outcomes for adults’ PUs managed in community settings?

What are the key nutritional factors that influence the prevention and treatment of PUs in patients cared for in community settings?

Is there a relationship between the nutritional status of patients and the frequency of development or healing of PUs in community settings?

The study application was structured utilizing the PICOS structure [30] prior to creating the search strings. The specific elements of the PICOS framework relevant to this study are outlined below:

Population (P): adults with PUs in community settings. 

This research targets patients with PUs who are receiving treatment in community environments, including nursing and care homes. This demographic is especially significant for exploring the impact of nutrition on PU care, given their regular exposure to varying levels of care and resource limitations.

Intervention (I): nutritional strategies. The primary intervention under examination consists of nutritional strategies applied in the management of ulcers. 

This encompasses the implementation of tailored diets, nutritional supplements, and specialized meal plans designed to facilitate the healing of PUs and enhance the nutritional status of patients in nursing or similar care environments. The SR will investigate different nutritional strategies, particularly those that emphasize proteins, vitamins, and essential minerals critical for skin health and tissue regeneration.

Comparison (C): nutritional strategies vs. no specific nutritional strategy. 

This SR will analyze the outcomes of patients who are following specific nutritional strategies versus those who are not. This comparison seeks to assess the effects of various nutritional approaches on the clinical results related to PUs. Furthermore, this study may evaluate different nutritional strategies to establish their relative effectiveness in PUs care.

Outcome (O): qualitative/quantitative outcomes. 

This study will assess qualitative and quantitative outcomes to offer a thorough understanding of the effectiveness of nutritional strategies. Quantitative outcomes may encompass the frequency and severity of PUs, healing rates, and changes in nutritional status (such as plasma protein and vitamin levels). Qualitative outcomes may involve thematic analyses of the experiences of patients and nursing staff related to the effectiveness of nutritional strategies in PU care.

Study Type (S): primary studies. 

This research will focus on primary studies that provide original data and findings relevant to the research question. The scope will include randomized controlled trials (RCTs), cohort studies, cross-sectional studies, and case-control studies. Using the PICOS framework, this research protocol seeks to systematically address the research question, ensuring a rigorous, comprehensive study design capable of producing meaningful and applicable results. This structured methodology will enable a thorough investigation into the effectiveness of nutritional strategies for managing PUs in community settings, ultimately providing valuable insights for nursing care and clinical practice.

### 2.4. Search Strategy

Prior to developing this SR, relevant international guidelines and existing SRs from the Cochrane Library will be examined to ensure alignment with this SR’s research aim [23,24]. Following this, searches will be conducted in the PubMed, Scopus, Embase, and CINAHL databases, complemented by an exploration of grey literature sources, including Google Scholar. The search strategy will utilize established terminology pertinent to the study topic and tailored strings specific to each database’s characteristics. Appropriate Boolean operators (AND/OR) will be applied to pinpoint studies that align with the research objective. Two researchers will independently perform a double-blind selection of articles, with a third researcher on hand to resolve any discrepancies. The free version of the citation management software Mendeley (Version 2.120.0) [31] will facilitate database compilation and duplicate elimination. To minimize selection bias, articles will undergo manual screening, and the bibliographies of included studies will be meticulously reviewed.

### 2.5. Query Search

#### 2.5.1. PubMed/Medline

((((((((“Diet” [Mesh] OR “Diet, Mediterranean” [Mesh] OR “Diet Therapy” [Mesh]) OR (“Diet, High-Protein” [Mesh] OR “Diet, Healthy” [Mesh] OR “Diet, Food, and Nutrition” [Mesh])) OR (“Nutritional Sciences” [Mesh] OR “Nutrition Therapy” [Mesh])) OR “Food” [Mesh]) OR (“Food, Fortified” [Mesh] OR “Food, Formulated” [Mesh])) OR “Dietary Supplements” [Mesh]) OR (“diet therapy” [Subheading] OR “Functional Food” [Mesh] OR “Probiotics” [Mesh] OR “Fish Oils” [Mesh])) AND (((((((((“Nurses” [Mesh] OR “Family Nurse Practitioners” [Mesh] OR “Nurse Practitioners” [Mesh] OR “Nurse Clinicians” [Mesh] OR “Nurse Specialists” [Mesh] OR “Nurses, Community Health” [Mesh]) OR “Evidence-Based Nursing” [Mesh]) OR “Advanced Practice Nursing” [Mesh]) OR “Family Support” [Mesh]) OR (“Nursing” [Mesh] OR “Home Health Nursing” [Mesh] OR “Family Nursing” [Mesh] OR “Rehabilitation Nursing” [Mesh])) OR (“Holistic Nursing” [Mesh] OR “Nursing Evaluation Research” [Mesh] OR “Clinical Nursing Research” [Mesh] OR “Nursing Research” [Mesh] OR “Skilled Nursing Facilities” [Mesh])) OR “Public Health Nursing” [Mesh]) OR (“Nursing Homes” [Mesh] OR “Nursing Care” [Mesh] OR “Home Nursing” [Mesh] OR “Geriatric Nursing” [Mesh])) OR “Community Health Nursing” [Mesh])) AND ((((((((((((((“Pressure Ulcer” [Mesh]) OR (pressure injury)) OR (pressure injuries)) OR (pressure injury prevention)) OR (pressure wound)) OR (pressure wound therapy)) OR (pressure sores)) OR (pressure sore prevention)) OR (pressure sores nursing)) OR (pressure sore)) OR (pressure ulceration)) OR (pressure ulcer treatment)) OR (pressure ulcer prevention)) OR (pressure ulcer prevention nursing)).

#### 2.5.2. Scopus

((TITLE-ABS-KEY (“Pressure Ulcer”) OR TITLE-ABS-KEY (“Pressure Injury”) OR TITLE-ABS-KEY (“Pressure Injuries”) OR TITLE-ABS-KEY (“Pressure Injury Prevention”) OR TITLE-ABS-KEY (“Pressure Wound”) OR TITLE-ABS-KEY (“Pressure Wound Therapy”) OR TITLE-ABS-KEY (“Pressure Sores”) OR TITLE-ABS-KEY (“Pressure Sore Prevention”) OR TITLE-ABS-KEY (“Pressure Sores Nursing”) OR TITLE-ABS-KEY (“Pressure Sore”) OR TITLE-ABS-KEY (“Pressure Ulceration”) OR TITLE-ABS-KEY (“Pressure Ulcer Treatment”) OR TITLE-ABS-KEY (“Pressure Ulcer Prevention”) OR TITLE-ABS-KEY (“Pressure Ulcer Prevention Nursing”))) AND ((TITLE-ABS-KEY (“Nurses”) OR TITLE-ABS-KEY (“Family Nurse Practitioners”) OR TITLE-ABS-KEY (“Nurse Practitioners”) OR TITLE-ABS-KEY (“Nurse Clinicians”) OR TITLE-ABS-KEY (“Nurse Specialists”) OR TITLE-ABS-KEY (“Nurse Community Health”) OR TITLE-ABS-KEY (“Evidence Based Nursing”) OR TITLE-ABS-KEY (“Advanced Practice Nursing”) OR TITLE-ABS-KEY (“Family Support”) OR TITLE-ABS-KEY (“Nursing”) OR TITLE-ABS-KEY (“Home Health Nursing”) OR TITLE-ABS-KEY (“Family Nursing”) OR TITLE-ABS-KEY (“Rehabilitation Nursing”) OR TITLE-ABS-KEY (“Holistic Nursing”) OR TITLE-ABS-KEY (“Nursing Evaluation Research”) OR TITLE-ABS-KEY (“Clinical Nursing Research”) OR TITLE-ABS-KEY (“Nursing Research”) OR TITLE-ABS-KEY (“Skilled Nursing Facilities”) OR TITLE-ABS-KEY (“Public Health Nursing”) OR TITLE-ABS-KEY (“Nursing Homes”) OR TITLE-ABS-KEY (“Nursing Care”) OR TITLE-ABS-KEY (“Home Nursing”) OR TITLE-ABS-KEY (“Geriatric Nursing”) OR TITLE-ABS-KEY (“Community Health Nursing”))) AND ((TITLE-ABS-KEY (“Diet”) OR TITLE-ABS-KEY (“Mediterranean Diet”) OR TITLE-ABS-KEY (“Diet Therapy”) OR TITLE-ABS-KEY (“Diet High Protein”) OR TITLE-ABS-KEY (“Diet Healthy”) OR TITLE-ABS-KEY (“Diet, Food and Nutrition”) OR TITLE-ABS-KEY (“Nutritional Sciences”) OR TITLE-ABS-KEY (“Nutrition Therapy”) OR TITLE-ABS-KEY (“Food”) OR TITLE-ABS-KEY (“Food, Fortified”) OR TITLE-ABS-KEY (“Food Formulated”) OR TITLE-ABS-KEY (“Dietary Supplements”) OR TITLE-ABS-KEY (“Functional Food”) OR TITLE-ABS-KEY (“Probiotics”) OR TITLE-ABS-KEY (“Fish Oils”))).

#### 2.5.3. CINAHL

(“diet Mediterranean” OR “diet therapy” OR “diet high-protein” OR “diet, healthy” OR (“diet, food, and nutrition”) OR “nutritional science” OR “nutrition therapy” OR “micronutrient supplementation” OR “food fortified with micronutrients” OR “food formulation” OR “dietary supplements” OR “functional food”) AND (nurse OR nurses OR “family nurse practitioner” OR “nurse practitioner” OR “nurse clinicians” OR “nurse specialist” OR “nursing education” OR “home health nursing” OR “family nursing” OR “rehabilitation nursing” OR “nursing home” OR “nursing care” OR “home nursing”) AND (“pressure ulcer” OR “pressure injury” OR “pressure injury prevention” OR “pressure wound” OR “pressure wound therapy” OR “pressure sore” OR “pressure sore prevention” OR “pressure sores nursing” OR “pressure ulcer treatment” OR “pressure ulcer prevention” OR “pressure ulcer prevention nursing” OR “pressure injuries”).

#### 2.5.4. Embase

(‘diet mediterranean’ OR ‘diet therapy’ OR ‘diet high-protein’ OR ‘diet, healthy’ OR ‘diet, food, and nutrition’ OR ‘nutritional science’ OR ‘nutrition therapy’ OR ‘micronutrient supplementation’ OR ‘food fortified with micronutrients’ OR ‘food formulation’ OR ‘dietary supplements’ OR ‘functional food’) AND (nurse OR nurses OR ‘family nurse practitioner’ OR ‘nurse practitioner’ OR ‘nurse clinicians’ OR ‘nurse specialist’ OR ‘nursing education’ OR ‘home health nursing’ OR ‘family nursing’ OR ‘rehabilitation nursing’ OR ‘nursing home’ OR ‘nursing care’ OR ‘home nursing’) AND (‘pressure ulcer’/exp OR ‘pressure ulcer’ OR ‘pressure injury’/exp OR ‘pressure injury’ OR ‘pressure injury prevention’ OR ‘pressure wound’ OR ‘pressure wound therapy’ OR ‘pressure sore’/exp OR ‘pressure sore’ OR ‘pressure sore prevention’ OR ‘pressure sores nursing’ OR ‘pressure ulcer treatment’ OR ‘pressure ulcer prevention’ OR ‘pressure ulcer prevention nursing’ OR ‘pressure injuries’).

### 2.6. Inclusion and Exclusion Criteria

This SR protocol establishes specific inclusion and exclusion criteria to ensure the homogeneity and reproducibility of the research that will be carried out (Table 1).

### 2.7. Quality Assessment and Risk of Bias Evaluation

In the article selection process, objectivity will be strictly upheld through a structured and systematic approach. Each publication will be independently assessed by two researchers using a double-blind methodology, ensuring that evaluations are made without influence from others. This methodological rigor is designed to eliminate subjective bias and enhance the credibility of the review process. To resolve any discrepancies arising from the independent evaluations, a third expert reviewer with extensive field experience will be appointed. This additional layer of review guarantees that decisions regarding article inclusion are fair, balanced, and based on a consensus of expert opinion.

The risk of bias and methodological quality of the included studies will be evaluated using the Critical Appraisal Skills Programme (CASP) checklists [32]. These tools provide a comprehensive framework for assessing the validity, relevance, and reliability of research findings. The CASP checklists enable a detailed examination of study design and methodology, focusing on aspects such as validity, which addresses the robustness and impartiality of the study design; relevance, which considers the applicability of findings to the research question; and results, which assess the clarity, reliability, and statistical integrity of reported outcomes. Each study will be thoroughly assessed based on the CASP criteria, with scores recorded in a standardized format to allow systematic comparison and synthesis.

The evaluation will encompass several critical domains: selection bias, which examines the representativeness of participant selection methods; performance bias, which assesses the influence of awareness of intervention allocations on outcomes; detection bias, which reviews the objectivity and blinding of outcome assessments; attrition bias, which considers the completeness and impact of dropout data on study results; and reporting bias, which ensures comprehensive and non-selective reporting of outcomes. By employing these rigorous evaluation tools, this study aims to meticulously assess the methodological quality and risk of bias in the included articles. This detailed and systematic approach is essential for enhancing the reliability and validity of the SR’s findings. Ultimately, this ensures that conclusions are based on high-quality evidence, contributing to more robust and trustworthy insights into the research question.

### 2.8. Data Extraction

To develop the SR, several critical elements will be meticulously identified and documented to ensure a comprehensive and detailed synthesis of the literature. This process will involve extracting essential information from each included study, which will be vital for subsequent data analysis and interpretation. Specifically, the following data will be extracted: Author(s), Year and Country of Study, Type of Study, Population, Setting, Intervention(s), Primary and Secondary Outcomes, and Results.

During this phase of the SR, a double-blind method will be employed to ensure objectivity and minimize bias in data extraction and management. Each researcher will independently extract and record data from the studies without knowledge of the other’s findings. This independent assessment is crucial for maintaining the integrity of the data collection process. To address any discrepancies that may arise, a third expert researcher will oversee the process, reviewing conflicting data points and making final decisions to reconcile differences, ensuring consistency and accuracy in the data included in the review.

Furthermore, a standardized data extraction form will be utilized to maintain uniformity and completeness in capturing all relevant information. This form will be pilot tested on a subset of studies to refine and standardize the extraction process. All extracted data will be entered into a comprehensive database designed for SRs, facilitating efficient data management, retrieval, and analysis.

By implementing these rigorous procedures, this SR aims to ensure a high level of methodological rigor and transparency. The meticulous identification and documentation of study elements, combined with a robust double-blind approach and expert supervision, will enhance the reliability and validity of this review’s findings. This thorough and systematic process is essential for generating robust and trustworthy evidence, ultimately contributing valuable insights to the field of research under investigation.

### 2.9. Data Synthesis

The included studies will be classified based on their primary and secondary objectives, ensuring precise alignment with this SR’s research aims. The extracted information will be presented in line with the original research. If substantial variability is detected among the studies, conducting a meta-analysis may not be appropriate. However, if applicable, the researchers will use the free-access software JASP version 0.19.0 [33]. JASP offers a user-friendly interface for conducting a range of statistical tests, including frequentist and Bayesian methods. It also allows for sensitivity analyses and assessments of publication bias, enhancing the reliability and validity of the findings. JASP’s comprehensive statistical outputs and visualizations facilitate clear data interpretation and presentation, ensuring well-supported conclusions.

## 3. Conclusions

This protocol aims to outline the methodology for conducting an SR on a critical topic of scientific and public interest. Given the importance of the subjects under investigation, a rigorous scientific approach is necessary to disseminate the most established knowledge regarding the management of PUs through nutritional interventions in elderly individuals within community settings. 

The primary objective of our study is to examine the effectiveness of nutritional strategies in the management of PUs among the elderly. Addressing our primary research question, an SR will be conducted to identify the impact of various nutritional interventions on the prevention and healing of PUs. Our results aim to demonstrate a significant correlation between appropriate nutritional support and improved outcomes in PUs management, highlighting the need for targeted nutritional strategies to enhance wound healing and overall QoL in the elderly. Secondly, the SR protocol focuses on how nutritional interventions can influence various aspects of PU management in elderly individuals, identifying key factors that contribute to better outcomes. 

Additionally, this protocol evaluates several predisposing factors for Pus’ development in the elderly, such as nutritional deficiencies and the quality of dietary intake. Comparing our results with previous primary studies, we anticipate observing a consistent pattern that underscores the positive effects of appropriate nutritional support on PU management. Consequently, our SR will contribute to this body of knowledge by providing a clearer understanding of effective nutritional strategies and their role in improving care for elderly individuals with PUs. We believe that addressing these issues requires the development of targeted interventions that promote optimal nutritional support. Educational programs aimed at raising awareness of the importance of nutrition in PU care and encouraging the implementation of evidence-based nutritional practices could be beneficial. 

Our SR protocol highlights the critical need for a comprehensive and methodologically sound approach to understanding the role of nutrition in managing PUs in elderly individuals within community settings. By systematically reviewing the existing literature, we can identify more effective strategies for preventing and managing PUs and improving overall patient outcomes. 

### Strengths and Limitations

The proposed SR aims to address existing gaps in the literature regarding the management of PUs in elderly individuals within community settings, with a particular focus on the role of nutritional interventions. The results are expected to provide new evidence on the correlations between nutritional status, PU outcomes, and overall QoL, contributing to the development of effective care strategies and improving the understanding of wound management in the elderly. A significant limitation in the development of this protocol may be the variability in the assessment and reporting of nutritional interventions and their impact on PUs. Other limitations to note in this research include the exclusive inclusion of studies published in English and the use of a limited selection of databases, although the focus was placed on the most relevant ones at the international level. Additionally, the highly specific search strategy of the protocol might lead to the exclusion of relevant studies and pose challenges in terms of reproducibility.

## Figures and Tables

**Table 1 diseases-12-00253-t001:** Inclusion and exclusion criteria.

Type of study	- Primary study - All other types of studies (e.g., editorials, commentaries, reviews, and protocol studies) will be excluded
Population	- Adult elderly people - Studies involving young and/or heterogeneous age populations will be excluded
Relevance	- Studies that are pertinent to the aim of this study
Temporal limit	- No limit
Language	- Primarily English - The authors will attempt to include studies in other languages (excluding Chinese) if relevant based on the abstract in English

## Data Availability

Data supporting this research are available in this manuscript and Appendix A.

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
