# Peer review of "The Role of Nutrition in the Nursing Management of Pressure Ulcers in Adult Community Settings: A Systematic Review Protocol"

_diseases, 2024, doi:10.3390/diseases12100253_

Round 1
Reviewer 1 Report
Comments and Suggestions for Authors
The study protocol "The Role of Nutrition in the Nursing Management of Pressure Ulcers in Community Settings: A Systematic Review Protocol" is dealing with very serious issue: chronic wounds care management.
Explain how important this protocol is!
Is there already a valid protocol for this issue?
Since the Percent match is big: 44%, look at that part (or whole manuscript) in more detail!
Author Response
Dear Reviewer,
thank you for your collaboration. In the file annx the comments suggested.
Best.
The Authors

Reviewer 2 Report
Comments and Suggestions for Authors
Thank you for this well-reported protocol on an important topic area. There are a few comments related to the abstract in the attached manuscript. I would also suggest being clear about how 'elderly' is being defined for the review.

Very well written overall, please see the commented manuscript for a few minor suggestions.
Author Response
Dear Reviewer,
thank you for your interest. The comments in the file annex.
Best.
The Authors

Reviewer 3 Report
Comments and Suggestions for Authors
Thank you for the opportunity to review your protocol titled The Role of Nutrition in the Nursing Management of Pressure Ulcers in Community Settings: A Systematic Review Protocol. It's an interesting protocol that tries to review the effect of various nutritional strategies in the nursing management of pressure ulcers in community settings. I have a few comments for the authors:
I recommend adding 'adult people' or similar to the title, because as you well know and indicate in the inclusion/exclusion criteria, you are going to limit your search to adults and not to children or adolescents, who may also suffer from them.
The objectives are well defined and feasible.
The databases consulted (Pubmed, CINAHL, Scopus) seem sufficient to achieve the proposed objectives.
Yoy should also consider one of the limitations of your protocol, such as language or the fact that it does not consult other databases (for example, LILACS). This aspect may limit access to relevant literature.
Author Response
Dear Reviewr,
thank you again for your collaboration. The comments in the annex file.
Best.
The Authors

Round 2
Reviewer 1 Report
Comments and Suggestions for Authors
After corrections, the manuscript "The Role of Nutrition in the Nursing Management of Pressure Ulcers in Adult Community Settings: A Systematic Review Protocol" was significantly improved. I propose publication in the Diseases Journal in its current form.